# Expression of IFN-Gamma is significantly reduced during severity of covid-19 infection in hospitalized patients

**Sajid Mansoor[1], Ayesha Raza Butt[2], Asima Bibi[2], Saima Mushtaq[3], Inayat Ullah[4], Fahad Alshahrani[5], Amjad Khan[2]\*, Atika Mansoor[6]\***

1 Department of Microbiology, Faculty of Science & Technology, University of Central Punjab, Lahore, Pakistan, 2 Department of Pharmacy, Quaid-i-Azam University, Islamabad, Pakistan, 3 Department of Healthcare Biotechnology, Atta-ur-Rahman School of Applied Biosciences (ASAB), National University of Sciences and Technology (NUST), Islamabad, Pakistan, 4 Department of Pediatrics, General Hospital, Islamabad, Pakistan, 5 Security Forces Hospital, Riyadh, Kingdom of Saudi Arabia, 6 Institute of Biomedical & Genetic Engineering (IB&GE), Islamabad, Pakistan

\* amjadkhan@qau.edu.pk (AK); atikamansoor@hotmail.com (AM)

**Data Availability Statement:** All relevant data are within the manuscript.

**Funding:** Recipient of award/grant: Atika Mansoor Grant # IBGE/IEC/17/02/21 Funder: Institute of

## Abstract

Cytokines play an important role in SARS-CoV-2 infection progression and severity. A number of inflammatory cytokines have been directly associated with disease severity including IL-6 (interleukin-6), IL-10, TNF-α (tumor necrosis factor alpha), IFN-γ (interferon-gamma). Here, in this study, the aim was to better understand the interplay between host immune response mediated by cytokines and severity of SARS-CoV-2 infection by assessing cytokine expression. Therefore, we measured expression levels of a total of 12 genes (IFNA-1, IFN-γ, IL-1α, IL-1β, IL-4, IL-6, IL-7, IL-10, IL-11, IL-13, IL-15, and IL-27) encoding inflammatory, anti-inflammatory and regulatory cytokines using QRT-PCR in hospitalized patients with severe infection compared to mildly infected. IFN-γ was identified as a potent marker of disease severity as indicated previously. Moreover, levels of IL-7 were also found to be partially reduced in patients compared to the healthy controls and linked negatively to disease severity. Identification of these cytokines may be helpful in not only understanding disease pathogenesis but also in better management of the patients after covid infection.

## Introduction

The novel beta-coronavirus, severe acute respiratory syndrome coronavirus 2 (SARS-CoV-2), that arose from China back in 2019, is now responsible for over 6.6 million deaths globally according to WHO (https://covid19.who.int/) [1]. Coronavirus family contains single stranded RNA viruses with a broad host range including humans [2]. Majority of the cases with SARS-CoV-2 show only mild to moderate symptoms but few, about 20%, progress towards acute bilateral pneumonia, a lethal condition, that may culminate at acute respiratory syndrome and multi-organ failure [3]. In humans the infection presents two imbricated stages starting with highly active virus replication activating human defenses in the form of a strong immune response [4]. The immune response is driven by multiple cytokines having diverse implications, few of these like IL-6 (interleukin-6), IL-10, TNF-α (tumor necrosis factor alpha), IFN-γ

Biomedical and Genetic Engineering (IBGE) URL:
https://hii-project.weebly.com/ The funders had no
role in study design, data collection and analysis,
decision to publish, or preparation of the
manuscript.

**Competing interests:** The authors have declared
that no competing interests exist.

(interferon-gamma) are viewed as independent markers of disease severity [5–8]. Moreover, an accelerated disease progression has been postulated to be directly related to an increase in inflammation, a dysfunctional host immune response culminating in a cytokine storm [9]. This postulate gets more strength by the fact that administration of anti-inflammatory steroids has shown to decrease the mortality rate by 29% in patients, however, the exact role of each of these cytokines is still indistinct to the severe stages of the disease. Furthermore, the significance of a dysfunctional immune response owing to cytokine dysregulation succumbing into a cytokine storm, is still not well understood [10].

Interferons have a vital role of restricting any viral infection and same is for the SARS-CoV-2, however, they are also responsible for disease progression towards a more severe form [11]. Interferon gamma (IFN-γ) is one of the key interferons to cope with viral infections as it narrows the viral replication by stimulating T lymphocytes for cytokine production along with an activation of the cytotoxic T lymphocytes [12]. A number of studies show that both, IFN-γ and IFN-β, downregulate the SARS-CoV-2 replication and also present synergistic antiviral role against it [13, 14]. Natural killer cells and T lymphocytes secrete the IFN-γ, and although it is a molecule driving adaptive immune system along with IL-4 and IL-6, still it has a vital role in all immune response phases [15]. Nevertheless, a persistent elevated level of IFN-γ aggravates the systemic inflammation contributing towards tissue injury and organ failure [16]. On the other hand, downregulation of IFN-γ is also a contributing factor towards the severity of infection as in the case of SARS-CoV-2.

Among the key interleukins, IL-6 has been identified as a major cytokine combating against the disease progression. In response to a variety of pathogens, especially the viruses, macrophages and T-lymphocytes produce this inflammatory interleukin, with great efficacy [17–19]. The IL-6 is a family of cytokines, comprising IL-6, IL-11, IL-27, IL-30, IL-31, oncostatin M (OSM), leukemia inhibitory factor (LIF), cardiotrophin-1 (CT-1), cardiotrophin like cytokine (CLC), ciliary neurotrophic factor (CNTF) and neuropoeitin (NP-1). Apart from IL-31, all other function through binding to a widely present β-receptor, gp130 (glycoprotein 130) [11]. This pleiotropic cytokine (IL-6) has dual effect pro-inflammatory as well as anti-inflammatory [20]. Only the homeostatic IL-6 concentration is effective in settling the tissue lesions and viral infections, on the other hand the excessive IL-6 value progresses towards cytokines storm [17–19]. A direct correlation has been established between the IL-6 and different stages of the disease also along with its radiological changes [6, 21–23]. Additionally, in different disease stages like a need for ventilator, or mortality or in both, the possible prognostic value has also been studied, when taken alone or in combination with other factors [24–28]. It has been reported that a decreased lymphocyte count, especially T-lymphocytes along with an increased level of IL-6 cytokine were associated with sever disease conditions [29, 30]. Where it is also suggested to monitor the IL-6 level throughout the disease progression for a better management of the disease [31].

Interleukin 1 (IL-1) plays a coordinating role in immune and inflammatory responses but also stimulates tissue repair and is one the most major pro-inflammatory cytokine. SARS-CoV-2 induced cytokine storm is due to an elevated level of these interleukins along with IL-6 and others [32]. The serum levels of IL-1 receptor antagonist (IL-1Ra) were significantly higher in patients, as well as among non-survivors when compared to surviving patients [33]. The imbalance between IL-1 and IL-1Ra levels can result in exaggerated inflammatory response and contribute to several human pathologies [34]. In COVID-19, the lung's epithelial injury leads to the secretion of IL-1α and the production of IL-1β. IL-1α recruit neutrophils to the infection site while IL-1β is a pro-inflammatory cytokine. Several studies reported high levels of IL-1β and IL-1Ra in the peripheral blood and bronchoalveolar lavage fluid of COVID-19 patients [35–38].

In addition, IL-13, an inflammatory cytokine, plasma titer is also a pronounced risk factor driving the patient status towards need of the mechanical ventilation [39]. This is a fatal stage and may progress towards multi organ failure even more drastic in the form of patient death [40, 41]. IL-13 is an immune regulatory cytokine which is essentially produced by activated helper T-lymphocytes (Th2) [42, 43]. Apart from Th2 cells a broad range of other immune cells also secret IL-13 including eosinophils, mast cells, basophils, natural killer cells, smooth muscle cells and fibroblast cells [44, 45]. It is reported that there is a functional association between IL-13 and IL-4 cytokines both also share almost 25% sequence homology and their genes are located on chromosome 5q31 [45]. A number of studies have provenan elevated IL-13 serum level in COVID patients where an increased level of IL-13in patients was associated with a requirement of mechanical ventilation support, more importantly patients receiving the IL-13 repression drugs showed much less severity of the disease [46].

In lieu of some clinical studies this has been extracted that the patients with more critical disease condition show lower counts of CD4[+] and CD8[+] cells while elevated levels of serum IL-10 and IL-6 in comparison to the patients with low or mild disease [47–49]. More importantly this lethal amalgam increases the mortality rate, advocating a pronounced role of IL-10 and IL-6 in viral pathogenesis [9]. Cytokines like IL-1, IL-6 and Tumor Necrosis Factor-α which are grouped in pro-inflammatory cytokines are more active in the first response to the disease while anti-inflammatory cytokine like IL-10 are more predominant in later stages when infection is sustained to count on the inflammation and to keep immune homeostasis [50]. IL-10 being a central molecule of the cytokine system downregulates the production of pro-inflammatory cytokines in the later stages of disease which aims at minimizing the damage induced by inflammatory cytokines [50, 51]. Furthermore, a meta-analysis research including 6242 patients also found increased levels of IL-10 and IL-6 in critical COVID patients [52]. Present study, therefore, was planned to explore further SARS-CoV-2 persuaded cytokines to better understand immune response mechanisms and more importantly if any potential therapeutic design against SARS-CoV-2 is possible involving cytokines.

## Methods

A total of 136 suspected SARS-CoV-2 infected patients, samples collected from Islamabad general teaching hospital, were included in the study. Out of these 68 were hospitalized, with SARS-CoV-2 confirmed either by covid PCR or HRCT while the other 68 were mild patients confirmed by covid PCR or by symptoms associated with covid. One hundred normal controls confirmed covid PCR negative samples were also collected for the study. Subjects were divided into following groups: 1. Severely Ill: Critically ill hospitalized patients, 2. Mildly Ill: Outdoor patients, who came to hospital with COVID-19 symptoms and later confirmed for COVID-19, 3. Control: Normal individuals with no underlying illnesses.

### Expression analysis

The role of various genes involved in innate and humoral immune response was investigated. Expression of a total of 12 genes (IFNA-1, IFNγ, IL-1α, IL-1β, IL-4, IL-6, IL-7, IL-10, IL-11, IL-13, IL-15, and IL-27) encoding inflammatory, anti-inflammatory and regulatory cytokines were analyzed using QRT-PCR. All primers sequences were obtained from GeneCard (https://www.genecards.org/). For the QRT-PCR, 5ml blood was collected from each patient and control and processed within one to two hours for RNA isolation. The blood samples were treated with lysis buffer and washed to remove RBCs from the peripheral blood lymphocytes. The PBLs pellet was used for RNA isolation by TRIzol-LS (Thermo Fisher Scientific Inc., Cat No: 10296–028) and cDNA was synthesized using RevertAidTranscriptase (ThermoScientificCat

No: EP0442) according to the manufacturer's instructions. GAPDH was used as house-keeping gene. Expression of selected target genes was measured by quantifying mRNA through QRT-PCR on SLAN-96P (Sansure Biotech). Each 20μl reaction mixture contained 10μl of 2X SYBRGreen/ROX qPCR Mix (Thermo FisherScientificCat No: K0221), 1μM of each forward and reverse primers, and 2μl of 1:10 cDNA. The PCR consisted of one cycle at 50˚C for 2 min, one cycle at 95˚C for 10 min and 45 cycles each of 95˚C for 15 sec and 60˚C for 1min with data acquisition at 60. Melting and dissociation curve was generated for each of the primer set to monitor amplification specificity. GAPDH gene was quantified as internal reference to calculate relative expression level of target gene. Relativistic expression (rE) of target gene was determined as: $rE = 100 \times 2^{-\Delta Ct}$, where $\Delta Ct$ = mean Ct (target gene) minus mean Ct (GAPDH).

## Statistical analysis

Statistical analysis was performed to assess the significance between variables using Microsoft Excel and SPSS tools and the relevant graphs were made on Prism by GraphPad. Values of $p$ below 0.1 were considered significant.

## Ethics statement

This study was conducted after getting ethical approval from the ethical review board of Institute of Biomedical and Genetic Engineering (Vide letter number: IBGE/IEC/17/02/21). Patients and controls were briefed about the study and informed written consent was obtained from all mild patients and controls. The severe patients mostly gave verbal consent as due to the covid imposed restrictions, the consent forms were not allowed inside the intensive care units and also many of them were not in a condition to sign the consent form. Detailed information for each patient was collected from hospital clinical notes. All procedures performed in studies involving human participants were in accordance with the ethical standards of the institutional and/or national research committee and with the 1964 Helsinki declaration and its later amendments or comparable ethical standards.

## Results and discussion

Considering SARS CoV-2 global health concern as a pandemic, we decided to explore and describe the human immune system behavior, in terms of specific cytokine blood levels through their expression. Our study focused on the levels of IFNA-1, IFNγ, IL-1α, IL-1β, IL-4, IL-6, IL-7, IL-10, IL-11, IL-13, IL-15 and IL-27at a given time in different groups of patients and healthy individuals that is expressed in detail in the underlines.

A total of 136 covid patients and 94 controls were included in the study which were initially divided into two groups based on the severity of the disease (Table 1). These patients were then further divided into four age groups where most of the patients fell into 45–64 years age group with the mean age of 50 years. However, when these patients were separated on the basis of disease severity (mild and severe), the mean age value for severe patients was 58.9 years (32 years-100 years). The mean age value for mild patients was 39.3 years (16 years -80 years). Among the patients 74% were males and 26% were females. The most common comorbidities were hypertension and diabetes. Others included cardiac disease, kidney disease, asthma, epilepsy and some psychological disorders.

Presence of comorbidity was strongly associated with disease severity especially diabetes and hypertension (Fig 1). The blood samples collected from these patients and controls were used to evaluate expression levels of 12 cytokines (Figs 2 and 3). The cytokines were selected on the basis of their role in response to the type of infection and the specific role played by the cytokine whether pro- or anti-inflammatory or regulatory.

**Table 1. Socio-demographic characteristics of study participants (n = 230).**

| | | Severe Patients | Mild Patients | Controls | Total |
|---|---|---|---|---|---|
| | | n = 68 (%) | n = 68 (%) | n = 94 (%) | n = 230 (%) |
| Age | <25 | 0 (0.0) | 7 (10.3 | 15 (16.0) | 22 (9.6) |
| | 25–44 | 13 (19.1) | 34 (50) | 53 (56.4) | 100 (43.5) |
| | 45–64 | 34 (50) | 22 (32.4) | 25 (26.6) | 81 (35.2) |
| | 65< | 21 (30.9) | 5 (7.3) | 1 (1.0) | 27 (11.7) |
| Gender | Male | 42 (61.8) | 59 (86.8) | 77 (81.9) | 178 (77.4) |
| | Female | 26 (38.2) | 9 (13.2) | 17 (18.1) | 52 (22.6) |
| Comorbidity | CD-HT+ | 3 (4.4) | 3 (4.4) | 1 (1.1) | 7 (3.0) |
| | CD-HT- | 2 (2.9) | 2 (2.9) | 1 (1.1) | 5 (2.2) |
| | Diabetes-HT+ | 11 (16.2) | 2 (2.9) | 0 (0.0) | 13 (5.7) |
| | Diabetes-HT- | 9 (13.2) | 3 (4.4) | 3 (3.2) | 15 (6.5) |
| | HT+-OM | 10 (14.7) | 3 (4.4) | 8 (8.5) | 21 (9.1) |
| | OM | 14 (20.6) | 8 (11.8) | 0 (0.0) | 22 (9.6) |
| | Nil | 19 (27.9) | 47 (69.1) | 81 (86.2) | 147 (63.9) |
| HTassociation | HT+-anymorbidity | 24 (35.3) | 8 (11.8) | 9 (9.6) | 41 (17.8) |
| | HT—anymorbidity | 25 (36.8) | 13 (19.1) | 4 (4.3) | 42 (18.3) |
| | Nil | 19 (27.9) | 47 (69.1) | 81 (86.2) | 147 (63.9) |
| Diabetassociation | Dia+anymorbidity | 20 (29.4) | 5 (7.4) | 3 (3.2) | 28 (12.2) |
| | Dia-antmorbidity | 29 (42.6) | 16 (23.5) | 10 (10.6) | 55 (23.9) |
| | Nil | 19 (27.9) | 47 (69.1) | 81 (86.2) | 147 (63.9) |

Levels of IFN-γ, IFN-α, IL-6, and IL-7 were found to be significantly related with SARS CoV-2 infection (Figs 2 and 3). Interferons are one of the most diverse cytokines with Type I and II actively involved against viral infections. The type I interferons alpha and beta are expressed by a variety of cell types [53] and act as first line of defense against most cellular infections, whereas type II has only one type of the gamma interferon which is important against viral, bacterial and fungal infections [54, 55].

This cohort study mainly comprised of three groups: normal individuals, patients with mild COVID disease and patients with severe COVID illness. Four of the lot of cytokines checked, interferon gamma (IFN-γ), interferon alpha (IFN-α), interleukin-6 (IL-6) and interleukin-7 (IL-7), were found to be significantly related to disease severity.

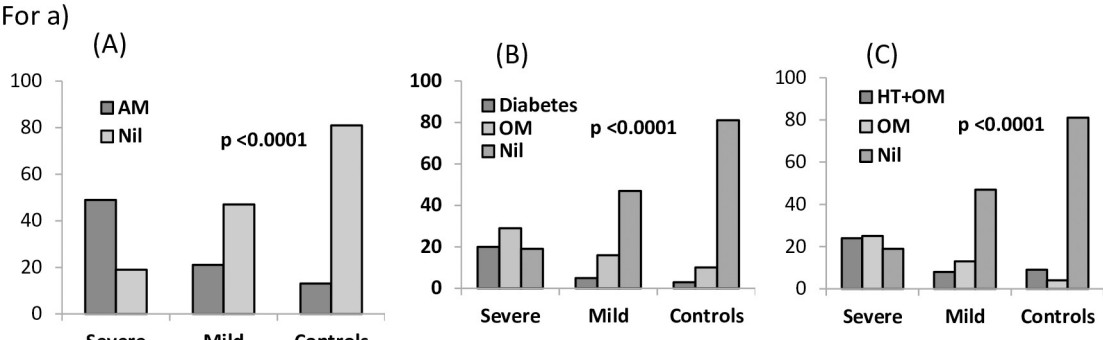

**Fig 1. Effect of co-morbidities on disease status.** (A) Any morbidity compared with no morbidity. (B)& (C) Diabetes and Hypertension association with disease severity. (Chi Sq Test). OM: Other morbidities.

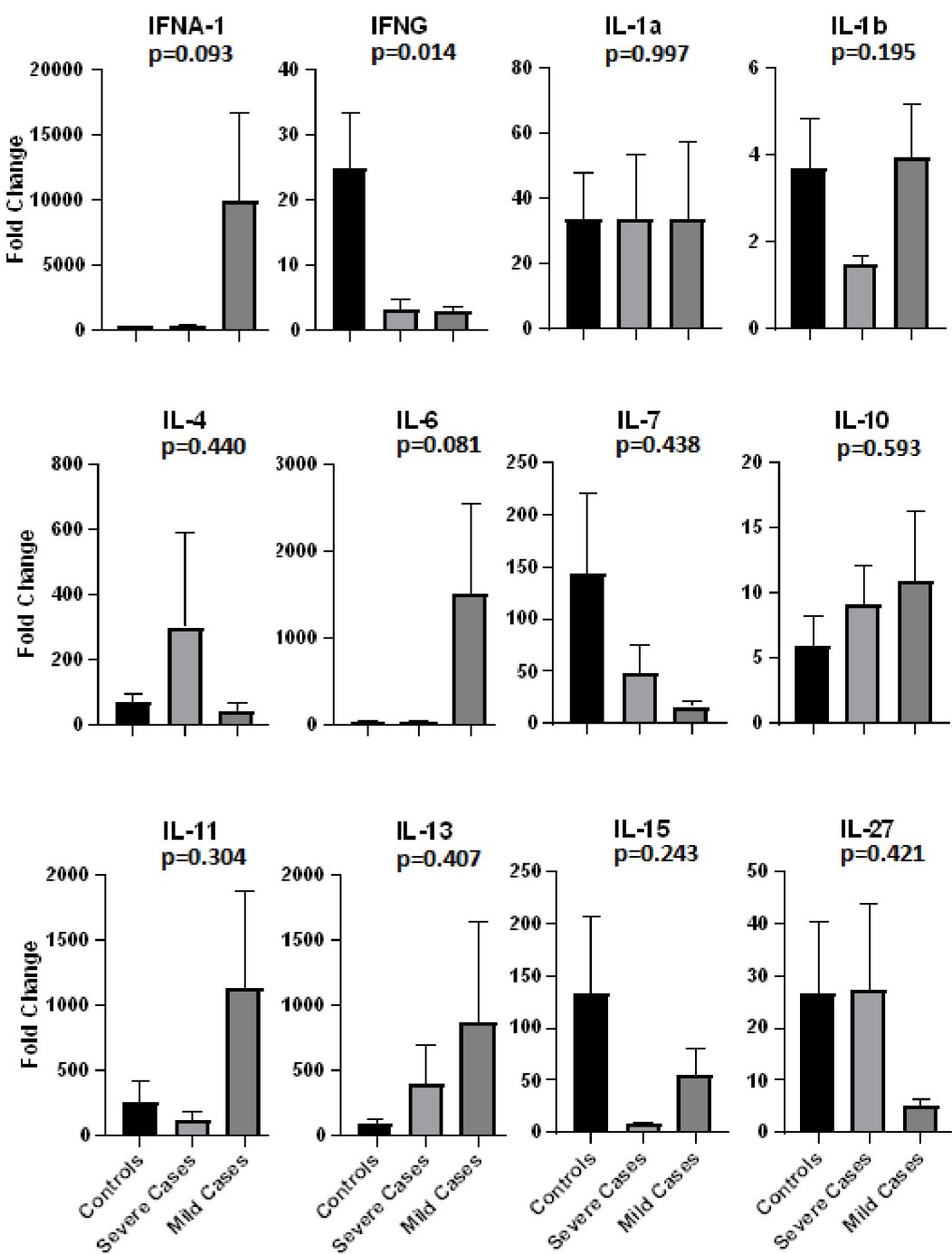

**Fig 2. Expression of various cytokines in covid severe and mild patients compared to controls.** (One way ANOVA).

Researchers around the globe have come up with much diverse cytokine and interferon results in COVID patients and the major factors driving this include difference in geographical location, demographic location, healthcare system, diseases stages, clinical settings and available resources [31]. In both mild and severely ill patient groups, the IFN-γ was found in significantly lower levels as compared to the normal individual group. Repeated sampling should be

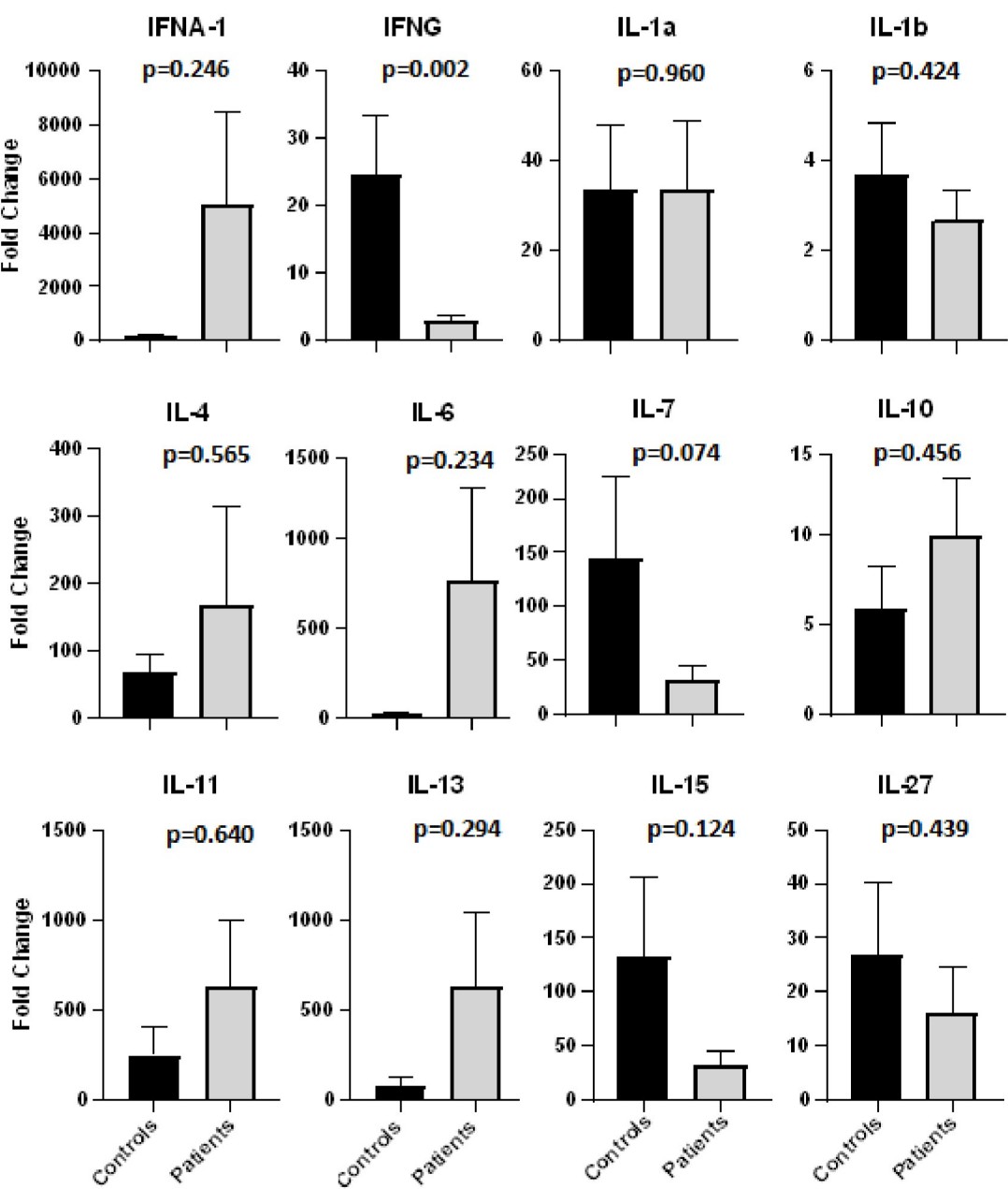

**Fig 3. Expression of various cytokines in covid patients compared to controls.** (T-Test).

done in order to build a complete panorama of cytokines and interferons levels, which unfortunately is difficult in current scenario, however even the results obtained with a single sample are also significant and relatable to other studies. According to *Sainz et al* (2004), previous studies focused on measuring the concentrations of IFN-β and -γ towards their viral replication inhibitory role, but his research proved that even a low concentration of IFN-γ and IFN-β (100U/ml) are quite effective in minimizing the SARS-CoV-2 virus replication to $1 \times 10^5$ – folds [14], which is quite in accordance to our results. These findings suggest that the IFNs based therapy should also be taken into consideration as a potential SARS-CoV-2 treatment.

Discussing about IL-6, we found relatively higher values in the mildly ill patients as compared to the normal as well as severely ill patients, however *Curz et al* reported the increasing value of IL-6 towards progression of disease to more severe stage [31]. Recently, a number of clinical studies have shown the contribution of IL-6 increasing systemic value towards severity of a number of viral infections including Andes virus [56], influenza virus [57], HBV [58], hepatitis C virus [59], human immunodeficiency virus [60], Crimean-Congo hemorrhagic fever virus [61] and Chikungunya virus [62]. Perhaps, certain viruses use this upregulation of IL-6 as a tool to escape the immune system, however the exact mechanism of IL-6 to viral infection response is still to be elucidated [17].

IL-7 level analysis of all the different patients in both mild and severe stages of the disease showed slightly higher levels in severely ill patients, which is in accordance with *Huang et al*, who reported elevated levels of IL-7 in ICU patients as compared to non-ICU patients [49]. In another study, however, the increased IL-7 levels corresponded to absolute number of T-cells independent of the severity of diseases [63]. Another research reported an increased level in IL-7 for both symptomatic and asymptomatic SARS-CoV-2 patients when compared to the healthy individuals, moreover male patients showed higher values of IL-7 as compared to female patients [64]. Furthermore, in different clinical trials the patients receiving IL-7 based treatment turned up with elevated lymphocyte count without any prove of hyper-inflammation and lung damage [65]. Still another study reported recovery of lymphopenic patient to increased lymphocyte cell count and normal interferons levels with also improvement in diseases status from sever to mild one [66]. These findings do tentatively suggest importance of IL-7 as a disease severity biomarker and also as a potential therapeutic.

Host immune system has a crucial role during SARS-CoV-2 pathogenesis. Interplay between different cytokines determine the outcome and severity of the infection. Understanding these may result in a better understanding of not only how these cytokines contribute towards pathogenesis but also if they can indicate disease progression or be used therapeutically. In this study we identify IFN-γ as a potent marker of disease severity as indicated previously. Moreover, levels of IL-7 were found to be reduced in patients compared to the healthy controls and linked negatively to disease severity. Finally, the study tries to provide a picture of levels of different cytokines in patients with severe to mild infection and may enhance our understanding about involvement of cytokines in SARS-CoV-2 pathogenesis.

## Conclusions

The most important goal of this study was to understand the role of cytokines in COVID-19 patients. Patients with SARS-CoV-2 infection had high levels of various cytokines, specifically IFN-γ was identified as a potent marker of disease severity that can be identified as an indicator of disease progression and a therapeutic goal.

## Limitations

we suggest that the exact timing for application of IFNγ-based therapeutics could be crucial: it should be earlier to significantly reduce the viral load and thus decrease the overall severity of the disease. In the field of drug and treatment, more activities should be done to find a solution to control and replicate the COVID-19 virus and ultimately reduce its side effects in clinics.

## Acknowledgments

We are grateful to the patients who participated in this study.

## Author Contributions

**Conceptualization:** Sajid Mansoor, Ayesha Raza Butt, Asima Bibi, Saima Mushtaq, Inayat Ullah, Amjad Khan, Atika Mansoor.

**Data curation:** Ayesha Raza Butt, Asima Bibi.

**Formal analysis:** Sajid Mansoor, Atika Mansoor.

**Investigation:** Ayesha Raza Butt, Asima Bibi, Inayat Ullah.

**Methodology:** Ayesha Raza Butt, Asima Bibi.

**Project administration:** Atika Mansoor.

**Resources:** Atika Mansoor.

**Supervision:** Amjad Khan, Atika Mansoor.

**Writing – original draft:** Sajid Mansoor.

**Writing – review & editing:** Saima Mushtaq, Fahad Alshahrani, Amjad Khan, Atika Mansoor.

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
