## [Decision Letter · Decision Letter 0]

17 Apr 2023

PONE-D-23-05389“Importance of IFN-Gamma and IL-7 in covid infection and cure”PLOS ONE

Dear Dr. Khan,

Thank you for submitting your manuscript to PLOS ONE. After careful consideration, we feel that it has merit but does not fully meet PLOS ONE’s publication criteria as it currently stands. Therefore, we invite you to submit a revised version of the manuscript that addresses the points raised during the review process.

We look forward to receiving your revised manuscript.

Kind regards,

Md Bashir Uddin, PhD

Academic Editor

PLOS ONE

4. Please include your tables as part of your main manuscript and remove the individual files. Please note that supplementary tables (should remain/ be uploaded) as separate "supporting information" files

Reviewers' comments:

Reviewer's Responses to Questions

**Comments to the Author**

1. Is the manuscript technically sound, and do the data support the conclusions?

Reviewer #1: Partly

Reviewer #2: No

2. Has the statistical analysis been performed appropriately and rigorously? 

Reviewer #1: Yes

Reviewer #2: Yes

3. Have the authors made all data underlying the findings in their manuscript fully available?

Reviewer #1: No

Reviewer #2: Yes

4. Is the manuscript presented in an intelligible fashion and written in standard English?

Reviewer #1: Yes

Reviewer #2: Yes

5. Review Comments to the Author

Reviewer #1: In the manuscript entitled, “Importance of IFN-Gamma and IL-7 in covid infection and cure.”, Monsoor et al have measured several cytokines in COVID-19 patients. They stratified the patients among severe, mild, or no illness groups and tried to find correlations between disease severity and measured cytokine expression. They also measured correlation between comorbidities and disease severity. In their cytokine measurements, they found a consistent decrease in IFNG expression. Expression of other cytokines tested were not significantly different between disease and no disease subjects. IFNG expression being down contrasts with several studies that used ELISA to measure its level in serum and found it be higher in more severe cases. This raises a question about study design. I have three major concern and one minor that I have listed below.

Major points

1. How were subjects recruited? Did normal control subjects come to hospital showing COVID-19 like symptoms and were found to negative for SARS-CoV2 infection?

2. Were normal subjects tested for unrelated infections or had symptoms?

3. How did authors normalize for difference in white blood cells in the blood?

Minor point

1. In figure 1, authors need to describe what “OM” means?

Reviewer #2: Manosoor et al. submitted the manuscript "Improtance of IFN-g and IL-7 in COVID infection and cure."

The authors examined the expression levels of intersted genes in blood and discovered the unique expression of IL-7 and IFN-g.

Nonetheless, publishing of the findings is premature at this point.

1. Because, as shown in Figure 2, there is a large drop in IFN-g expression in both severe and moderate instances, which is quite common in most viral infections. It will not add anything new to the research.

2. The amount of IL-7 expression in figure 3 is not particularly significant, which might be an artifact. However, the evidence should be validated by reproducible cell-based or mouse models.

As a result, I recommended that the manuscript be rejected.

6. PLOS authors have the option to publish the peer review history of their article (what does this mean?). If published, this will include your full peer review and any attached files.

Reviewer #1: No

Reviewer #2: **Yes: **Srinivasa Reddy Bonam

---

## [Author Response · Author response to Decision Letter 0]

11 May 2023

Dr. Bashir Uddin,

Academic Editor,

PLOS ONE, Dated: May 10, 2023

Subject: Rebuttal; Response to Reviewers

Dear Editor,

I hope you are well.

We have tried our best to respond to the comments by the esteemed reviewers, mentioned below. The responses are in light blue. Suitable changes have also been done in the main text (attached along as marked file). The study in our humble view has its merits to be published in your prestigious journal.

Reviewer #1:

In the manuscript entitled, “Importance of IFN-Gamma and IL-7 in covid infection and cure.”, Monsoor et al have measured several cytokines in COVID-19 patients. They stratified the patients among severe, mild, or no illness groups and tried to find correlations between disease severity and measured cytokine expression. They also measured correlation between comorbidities and disease severity. In their cytokine measurements, they found a consistent decrease in IFNG expression. Expression of other cytokines tested were not significantly different between disease and no disease subjects. IFNG expression being down contrasts with several studies that used ELISA to measure its level in serum and found it be higher in more severe cases. This raises a question about study design. I have three major concern and one minor that I have listed below.

- Response to the highlighted comment from the esteemed reviewer: As exact transcript from an article

Title: Alterations in the Expression of IFN Lambda, IFN Gamma and Toll-like Receptors in Severe COVID-19 Patients – DOI:https://doi.org/10.3390/microorganisms11030689 – Transcript: “With regard to the analysis of IFNγ levels in COVID-19 patients, transcript amounts for IFNγ were reduced in the lower respiratory tracts and PBMCs of critically ill patients compared to healthy donors despite all patients had negative results for plasma anti-IFNγ autoantibody detection. In agreement with these findings, a previous study showed that lower levels of IFNγ were associated with greater COVID-19 severity [48]. However, these results are in contrast with further previous findings, in which IFNγ protein levels in blood were upregulated in severe COVID-19 patients, contributing to a “cytokine storm” [24,49]. Possible explanations for this discrepancy are the following: i) a lower number of T lymphocytes expressing IFNγ have been reported in the blood of convalescent COVID-19 patients compared to healthy donors [50]; ii) increased frequencies of exhausted NK cells expressing lower levels of IFNγ have been described in COVID-19 patients [51]; iii) the antiviral transcriptional response in circulating immune cells has been shown to be strongly associated with a specific subset of IFNs, most prominently IFNα2 and IFNγ, and differential IFN subtype production was linked to distinct circulating immune cell types [52].”

All of the above mentioned reasons may contribute towards a depiction of lowered final interferon gamma transcript levels compared to its protein levels, as indicated by several analyses as well as ours. 

References (mentioned above):

24. Karki, R.; Sharma, B.R.; Tuladhar, S.; Williams, E.P.; Zalduondo, L.; Samir, P.; Zheng, M.; Sundaram, B.; Banoth, B.; Malireddi, R.K.S.; et al. Synergism of TNF-α and IFN-γ Triggers Inflammatory Cell Death, Tissue Damage, and Mortality in SARS-CoV-2 Infection and Cytokine Shock Syndromes. Cell 2021, 184, 149–168.e17.

48.Cremoni, M.; Allouche, J.; Graça, D.; Zorzi, K.; Fernandez, C.; Teisseyre, M.; Benzaken, S.; Ruetsch-Chelli, C.; Esnault, V.L.M.; Dellamonica, J.; et al. Low baseline IFN-γ response could predict hospitalization in COVID-19 patients. Front. Immunol. 2022, 13, 953502.

49.Costela-Ruiz, V.J.; Illescas-Montes, R.; Puerta-Puerta, J.M.; Ruiz, C.; Melguizo-Rodríguez, L. SARS-CoV-2 infection: The role of cytokines in COVID-19 disease. Cytokine Growth Factor Rev. 2020, 54, 62–75.

50. Yang, J.; Zhong, M.; Zhang, E.; Hong, K.; Yang, Q.; Zhou, D.; Xia, J.; Chen, Y.Q.; Sun, M.; Zhao, B.; et al. Broad phenotypic alterations and potential dysfunction of lymphocytes in individuals clinically recovered from COVID-19. J. Mol. Cell Biol. 2021, 13, 197–209.

51.Varchetta, S.; Mele, D.; Oliviero, B.; Mantovani, S.; Ludovisi, S.; Cerino, A.; Bruno, R.; Castelli, A.; Mosconi, M.; Vecchia, M.; et al. Unique immunological profile in patients with COVID-19. Cell. Mol. Immunol. 2021, 18, 604–612.

52. Galbraith, M.D.; Kinning, K.T.; Sullivan, K.D.; Araya, P.; Smith, K.P.; Granrath, R.E.; Shaw, J.R.; Baxter, R.; Jordan, K.R.; Russell, S.; et al. Specialized interferon action in COVID-19. Proc. Natl. Acad. Sci. USA 2022, 119, e2116730119.

Major points by Reviewer #1:

1. How were subjects recruited? Did normal control subjects come to hospital showing COVID-19 like symptoms and were found to negative for SARS-CoV2 infection?

Response: Subjects were divided into following groups: 1. Severely Ill: Critically ill hospitalized patients confirmed for COVID-19, 2. Mildly Ill: Outdoor patients, who came to hospital with COVID-19 symptoms and later confirmed for COVID-19, 3.Control: Normal individuals with no underlying illnesses. All patients that were either hospitalized or came to OPD between 11-02-2021 to 08-08-2021 were included for the study. As for the normal control subjects, no, they did not come to hospital with COVID-19 symptoms, rather, normal individuals without any illness or morbidity were taken as control. 

2. Were normal subjects tested for unrelated infections or had symptoms?

Response: The normal subjects had no symptoms but were tested for COVID-19 and were confirmed COVID-19 negative.

3. How did authors normalize for difference in white blood cells in the blood?

Response: Since we used quantitative PCR for the estimation of expression, a house keeping gene (GAPDH) was used to normalize the final results. Basically, an equal amount of RNA was used for cDNA synthesis for each sample and then GAPDH Ct values for all samples were normalized that in turn serves to point out any difference present for other genes being checked. Therefore, no normalization was performed at the white blood cells’ level, rather, equal amount of RNA and a house keeping gene was used.

Minor point by Reviewer #1:

1. In figure 1, authors need to describe what “OM” means?

In figure 1, “OM” refers to “Other Morbidities”. Needful is done. 

-----

Reviewer #2:

Mansoor et al. submitted the manuscript "Importance of IFN-g and IL-7 in COVID infection and cure."The authors examined the expression levels of interested genes in blood and discovered the unique expression of IL-7 and IFN-g. Nonetheless, publishing of the findings is premature at this point. 1. Because, as shown in Figure 2, there is a large drop in IFN-g expression in both severe and moderate instances, which is quite common in most viral infections. It will not add anything new to the research.

Response: The study does have its merit being a report from our population while still being relevant to the larger context by substantiating previous findings. Altogether, the finding here will contribute to the existing body of knowledge.

2. The amount of IL-7 expression in figure 3 is not particularly significant, which might be an artifact. However, the evidence should be validated by reproducible cell-based or mouse models.

Response: The title has been changed and now the main emphasis is on interferon gamma instead of IL-7. Similar changes have been made in the main body of the manuscript text.

Please do extent any queries. Take care.

Kind Regards,

Research Team

Amjad Khan, PhD (Corresponding Author)

E-mail: amjadkhan@qau.edu.pk

---

## [Editor Report · Decision Letter 1]

10 Aug 2023

PONE-D-23-05389R1

"Expression of IFN-Gamma is significantly reduced during severity of covid-19 infection in hospitalized patients"

PLOS ONE

Dear Dr. Khan,

Thank you for submitting your manuscript to PLOS ONE. After careful consideration, we feel that it has merit but does not fully meet PLOS ONE’s publication criteria as it currently stands. Therefore, we invite you to submit a revised version of the manuscript that addresses the points raised during the review process.

Specifically, author addressed almost all the reviewers questions. However, question no. 2 from reviewer 2 partially answered by authors. Authors need to be address that question.

We look forward to receiving your revised manuscript.

Kind regards,

Md Bashir Uddin, PhD

Academic Editor

PLOS ONE
---

## [Author Response · Author response to Decision Letter 1]

16 Aug 2023

Dr. Bashir Uddin,

Academic Editor,

PLOS ONE, Dated: 15-08-2023

Subject: Rebuttal; Response to Reviewers

Dear Sir,

We are extremely pleased regarding the review that only a minor revision remains. The 2nd comment by the esteemed reviewer 2, as pointed out by the respected academic editor along with an appropriate response is as following:

Comment # 2 by Reviewer # 2:

The amount of IL-7 expression in figure 3 is not particularly significant, which might be an artifact. However, the evidence should be validated by reproducible cell-based or mouse models.

Response:

The title has been changed and now the main emphasis is on interferon gamma instead of IL-7. Similar changes have been made in the main body of the text.

However, as for the second part of the comment by reviewer 2, any further validation about whether the IL-7 expression is actual or just an artifact through reproducible cell-based or mouse models, warrants a separate study. We agree with the esteemed reviewer that it might just be an artifact, nonetheless, we report it as per the data collected. Moreover, we try to make no extended claims based on this and have corrected anything as such within the version of the manuscript submitted.

We sincerely hope that this clarifies any ambiguities and makes the manuscript appropriate for acceptance.

Thank you.

Please do extent any queries. Take care.

Best regards

Dr. Amjad Khan

Corresponding author

Email: amjadkhan@qau.edu.pk

---

## [Editor Report · Decision Letter 2]

29 Aug 2023

"Expression of IFN-Gamma is significantly reduced during severity of covid-19 infection in hospitalized patients"

PONE-D-23-05389R2

Dear Dr. Khan,

We’re pleased to inform you that your manuscript has been judged scientifically suitable for publication and will be formally accepted for publication once it meets all outstanding technical requirements.

Kind regards,

Md Bashir Uddin, PhD

Academic Editor

PLOS ONE
---

## [Editor Report · Acceptance letter]

18 Sep 2023

PONE-D-23-05389R2 

Expression of IFN-Gamma is significantly reduced during severity of covid-19 infection in hospitalized patients 

Dear Dr. Khan:

I'm pleased to inform you that your manuscript has been deemed suitable for publication in PLOS ONE. Congratulations! Your manuscript is now with our production department. 

Kind regards, 

on behalf of

Dr. Md Bashir Uddin 

Academic Editor

PLOS ONE